# Position: Is Current Research on Adversarial Robustness Addressing the Right Problem?

## Abstract

Short answer: Yes, Long answer: No! Indeed, research on adversarial robustness has led to invaluable insights helping us understand and explore different aspects of the problem. Many attacks and defenses have been proposed over the last couple of years. The problem, however, remains largely unsolved and poorly understood. Here, I argue that the current formulation of the problem serves short term goals, and needs to be revised for us to achieve bigger gains. Specifically, the bound on perturbation has created a somewhat contrived setting and needs to be relaxed. This has misled us to focus on model classes that are not expressive enough to begin with. Instead, inspired by human vision and the fact that we rely more on robust features such as shape, vertices, and foreground objects than non-robust features such as texture, efforts should be steered towards looking for significantly different classes of models. Maybe instead of narrowing down on imperceptible adversarial perturbations, we should attack a more general problem which is finding architectures that are simultaneously robust to perceptible perturbations, geometric transformations (*e.g.* rotation, scaling), image distortions (lighting, blur), and more (*e.g.* occlusion, shadow). Only then we may be able to solve the problem of adversarial vulnerability.

## 1. Introduction

Adversarial vulnerability is the Achilles heel of deep learning. A small imperceptible perturbation is enough to fool a neural network (Szegedy et al., 2014; Goodfellow et al., 2015) (See Fig. 1.a). A sample $x'$ is said to be an adversarial

example for $x$ when $x'$ is close to $x$ under a specific distance metric, while $f(x') \neq y$. Formally:

$$x' : D(x, x') < \epsilon \ \text{ s.t } \ f(x') \neq y \tag{1}$$

where $D(.,.)$ is a distance metric, $\epsilon$ is a predefined distance constraint (*a.k.a* allowed perturbation), $f(.)$ is a neural network, and $y$ is the true label of sample $x$ (*i.e.* oracle $g(x)$).

Significant efforts have been devoted to solving this problem since the resurgence of neural networks. Not much progress, however, has been made. Surprisingly, adversarial training[1] or its variations thereof, proposed by those who first popularized the problem (Goodfellow *et al.* (Goodfellow et al., 2015)), remains the most effective solution. Here, without going much into details, I argue that the current formulation of the problem, while being useful, is misleading. I also discuss some aspects that are critical for solving this problem and list what I consider might be good directions to explore. Please note that this piece is entirely my personal take on the topic, hoping it will spark further discussions.

**Here, we criticize the prevailing focus on imperceptible adversarial perturbations and suggest that this narrow scope may hinder progress in developing truly robust models.**

## 2. Discussion

Following, I highlight some discrepancies and misconceptions pertaining to adversarial examples and human visual perception. My principal focus is on deep neural networks (DNNs) for vision, however, a lot of the arguments also apply to other domains.

### 2.1. How practical is the current formulation?

A system should be robust regardless of perturbations being imperceptible or not. Take perturbation of a traffic sign in the context of self-driving vehicles as an example. An adversary may replace the pristine sign with a perturbed one to mislead the vehicle. Why does the perturbed image have to be imperceptible? A possible answer is *because otherwise*

[1]Anonymous Institution, Anonymous City, Anonymous Region, Anonymous Country. Correspondence to: Anonymous Author <anon.email@domain.com>.

Preliminary work. Under review by the International Conference on Machine Learning (ICML). Do not distribute.

---

[1]Adding adversarial examples to the training set and retraining the model, at the cost of computational overhead and reduced accuracy.

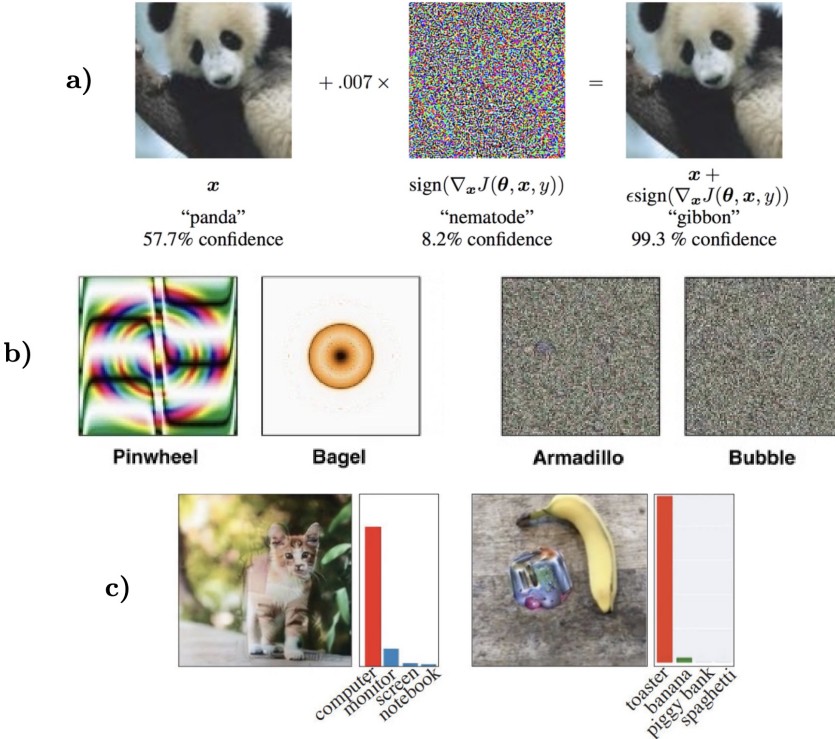

Figure 1: a) An adversarial example generated for the giant panda image using the FGSM attack (Goodfellow et al., 2015). b) Example meaningless patterns (*a.k.a* fooling images) that are classified as familiar objects by a DNN (Nguyen et al., 2015). c) Examples of invisible (left) and visible (right) backdoor attacks (*e.g.* (Brown et al., 2017)). Placing a "sticker" next to a banana can fool a CNN into classifying the image of the banana as a toaster.

*someone will notice and report it*! However, this is not a good reason since we can not always have a person to police the input, especially for models that are supposed to run at scale. The adversary may always be able to find an opportunity to submit a perceptible malicious query (*e.g.* when no one is watching). Thus, there is no point in assuming imperceptibility. Further, the adversary may post a natural-looking fooling image (*e.g.* in form of an Ad, or a physical sticker (Eykholt et al., 2018; Sharif et al., 2019)) to fool the system (Fig. 1.b). As another example, consider a system that processes millions of images to perform face verification. Assuming a person in the loop to check the input faces defies the purpose of building fully automated verification systems. On one hand, maybe the reason why we are infatuated by the current formulation is that it does not feel right when a model makes a clear mistake, while the scene has almost not changed for us. On the other hand, errors induced by perceptible perturbations are equally important as errors induced by imperceptible perturbations, and perhaps are more prevalent.

Nevertheless, adversarial attacks are useful in some applications and for some specific purposes (*e.g.* for system identification (Borji, 2020)). A type of attack, known as backdoor

or trojan attack[2], seems to be more of a threat than imperceptible adversarial attacks, and hence is more practical (Fig. 1.c).

## 2.2. Bounded perturbation assumption is restrictive and misleading

The current formulation of the problem is somewhat contrived. It has led us to fixate on the wrong, or at best, incomplete class of models. It is possible that adversarial vulnerability does not have any solution using existing DNN architectures. Thus, we may have to explore a bigger hypothesis space[3] in which models offer robustness to larger variations and perturbations (*e.g.* similar to the human visual system). In general, $x'$ is an adversarial example for $x$ if it forces the model $f(.)$ to make a mistake (preferably with high confidence)[4]:

$$x' : \left[ g(x') = g(x) \ \wedge \ f(x') \neq g(x) \right] \quad (2)$$

---

[2]Some malicious examples are planted in the training data to fool the model.

[3]Current models are limited in a sense that they all use the same building blocks. Current tools and software also exacerbate the issue and make it harder to think out of the box.

[4]This formulation is for non-targeted attacks. It can be easily modified to cover targeted attacks.

Here, we assume that the initial prediction for sample $x$ is correct (*i.e.* $f(x) = g(x)$). The formulation in Eq. 2 covers other types of class-preserving transformations as well, including translation, rotation, scaling, Gaussian blur, etc. It also subsumes Eq. 1, and considers a larger fraction of perturbations, with some being perceptible (Fig. 2.a). Notice that the hidden assumption in Eq. 1 is that $f(x) = y$, otherwise altering the input to fool the model would not make sense. By varying the magnitude of perturbation, here parameterized by $\theta$, a psychometric function[5] similar to the ones depicted in Fig. 2.b can be obtained. The model performance drops as the perturbation grows. According to this figure, model B is less robust than model A since its psychometric function falls below the psychometric function of model A.

According to Eq. 2, **whether the model response agrees with the human response is irrelevant. What matters is the agreement with the ground truth (oracle) response, which may be determined by human experts (or via accurate physical measurements; but not the average person). The goal is not to make a model behave exactly like humans (*i.e.* respond the same to all inputs). However, since human vision is very robust, it can guide us to discover robust features to strengthen our models** (will be elaborated in the next section). Since the oracle is determined by human experts, a model that is able to robustly predict the ground-truth labels will naturally behave similar to humans, and thus it will be insensitive to imperceptible adversarial perturbations.

Eq. 2 does not cover fooling images or samples that simply cause a model to fail:

$$x' : \Big[ f(x') \neq g(x') \Big] \tag{3}$$

Therefore, in addition to testing for robustness, models should be evaluated in terms of their accuracy over a wider range of inputs, some of which may be nonsensical[6].

The main purpose of extending the definition of robustness, as in Eq. 2, is to intentionally make the problem harder. This definition shrinks the space of possible models, since more types of perturbations are considered now.

### 2.3. $L_p-$norm is not a good measure of perceptual similarity

Even emphasizing the imperceptibility of perturbations, there are sill problems with Eq. 1. The $L_p$-norm assumption sits at the core of the current formulation. This assumption, however, is not always valid (Sharif et al., 2018). For ex-

ample, a slight rotation or translation of an image is almost indistinguishable to humans, but it can cause a big $L_p$-norm distance (Fig. 3). Conversely, an image can be manipulated to cause a big change in human perception with a small $L_p$-norm perturbation (*e.g.* removing some important edges or vertices in an image; or changing only one pixel). Further, there might be two images from two different categories that have very small $L_p$-norm distance. Although different types of $L_p$-norm have been used for adversarial robustness in the literature (*e.g.* $L_0, L_2, L_\infty$), no single $L_p$-norm can explain all aspects of perceptual similarity judgments by humans. The $L_p$-norm measure treats all the pixels the same, whereas we know some pixels are more important than the others, for example those forming the edges or salient regions (Borji & Itti, 2012)(more on this in the next subsection).

### 2.4. Robust *vs.* non-robust features

Current DNNs do not distinguish between robust and non-robust features[7]. This is also known as "shortcut learning"[8] where models latch on to any feature, even spurious correlations, to solve the task (Geirhos et al., 2020). Some important robust features are shape, edges, vertices, corners, object and surface boundaries, and Gestalt principles (Biederman, 1987; Pomerantz & Portillo, 2011). Humans rely heavily on these cues to recognize objects. Conversely, convolution-based models[9] are biased more towards texture (Geirhos et al., 2018a; Baker et al., 2018) (Fig. 4.a). Fooling images also attest to this (Fig. 1.b). Object shape remains largely invariant to imperceptible adversarial perturbations (Fig. 4.b). This is why adversarial examples seem so bizarre to humans. The convolution operation is biased towards capturing texture, since the number of pixels constituting texture far exceeds the number of pixels that fall on the object boundary. This in part explains why Convolutional Neural Networks (CNNs) are susceptible to adversarial examples. I suggest focusing on sketch recognition to build and test models that prioritize shape over texture in recognition. Adding a small perturbation to a sketch image would be easier to notice due to the lack of background texture (Fig. 4.c). Robust features will also help alleviate backdoor attacks. **In sum, a consensus is emerging centered around the hypothesis that adversarial vulnerability is due to deep models relying more on non-robust features that are predictive of class labels.**

---

[5]A term commonly used in cognitive sciences to describe the relationship between stimulus and response.

[6]Or the model should not be confident about its predictions when the oracle is not confident either.

[7]"Features derived from patterns in the data distribution that are highly predictive, yet brittle and (thus) incomprehensible to humans" (Ilyas et al., 2019).

[8]"Shortcuts are decision rules that perform well on standard benchmarks but fail to transfer to more challenging testing conditions" (Geirhos et al., 2020).

[9]Including CNNs, Capsule Networks (Sabour et al., 2017), Transformers (Dosovitskiy et al., 2020), and MLP-like models (Tolstikhin et al., 2021). Notice that the latter two types of models use convolution to perform patch embedding!

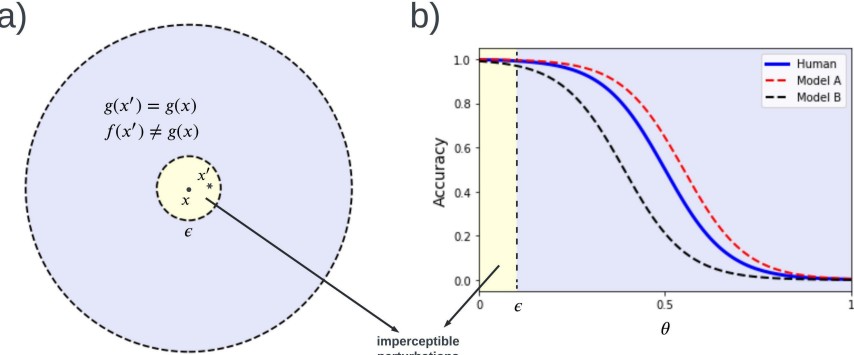

Figure 2: a) Illustration of adversarial robustness. a) Adversarial examples within the yellow $\epsilon$-ball are imperceptible and constitute a small fraction of possible perturbations. b) The hypothetical performance curves (*a.k.a* psychometric functions) of humans versus a more robust model (B) and versus a less robust model (A). The current definition (Eq. 1), encourages the models to be robust in the imperceptible region, and ignores the rest. A model is fully robust if it performs perfectly for all perturbation magnitudes (*i.e.* Accuracy=1 for all $\theta$s). In reality, however, the performance curves look like a reverse sigmoid function, indicating that performance drops as the perturbation grows.

**Existing deep architectures are not capable of detecting robust features and trying to fix the problem only with ML tricks (*e.g.* adversarial training), rather than looking for missing components or new classes of models, will not take us far.**

### 2.5. Adversarial robustness *vs.* generalization

The bulk of research on adversarial robustness has been loyal to the formulation in Eq. 1. They have primarily emphasized on using ML techniques to make existing DNNs robust. Few works have recently started to explore computations and building blocks beyond those employed in current architectures (*e.g.* background subtraction (Xiao et al., 2020; Moayeri et al., 2022; Borji, 2021)).

Some works have proposed that there is a trade-off between robustness and standard generalization (Tsipras et al., 2018), whereas others have argued the opposite (Stutz et al., 2019). Human vision is the existing proof that both are achievable at the same time. Note that almost all of our findings and intuitions on adversarial robustness are bound to the current class of models. Even neural architecture search methods (See (Elsken et al., 2019) for a review) are biased in the sense that they search in the space of models that use the same building blocks as the existing architectures. Thus, one path to solve the problem is to look for significantly different, or entirely new, classes of models that are invariant to both adversarial attacks and other variations. Overall, the problems of adversarial robustness and generalization are two sides of the same coin, and they should be studied simultaneously. The latter encompasses robustness to a wider range of variations and has deeper historical roots. To gain some perspective, let's recap some limitations, in addition to sensitivity to adversarial noise, of DNNs. They,

- are sensitive to image distortions such as blur, fog, Gaussian noise (Hendrycks & Dietterich, 2019), and filtering in the Fourier domain (Jo & Bengio, 2017),

- fail to detect objects embedded in novel contexts or occluded by out-of-context objects (*e.g.* elephant in the living room (Rosenfeld et al., 2018)),

- are, in contrast to common belief, surprisingly limited in terms of invariance to geometric transformations such as translation (Azulay & Weiss, 2019), rotation, and scaling. In fact, detection of small object in scenes still remains a major challenge,

- show near-perfect accuracy when trained on randomly shuffled image labels (Zhang et al., 2021), suggesting maybe they are just memorizing the input (*a.k.a* rot memorization),

- can handle specific kinds of noise when noisy images are incorporated in the training sets. They, however, fail to generalize to unseen noise patterns, even those similar to training patterns (Geirhos et al., 2018b),

- show weak out-of-distribution generalization (*e.g.* (Recht et al., 2019; Barbu et al., 2019; Borji, 2021)), suggesting that maybe we have overfitted ourselves to the existing benchmarks,

- have difficulty in learning some tasks such as same–different tasks even after being presented with millions of training examples (Fleuret et al., 2011; Ellis et al., 2015; Kim et al., 2018; Stabinger et al., 2021), and

Figure 3: Illustration of $L_p$-norms for different types of geometric transformations, image distortions, and adversarial perturbations over a sample image from the MNIST dataset.

- require large amounts of training data, and are prone to catastrophic forgetting.

### 2.6. Argument on visual illusions

To justify, or perhaps downplay, the problem of adversarial vulnerability, some people argue that human vision has also its own blind spots and can be fooled by visual illusions[10], in spite of evolution spending a lot of time optimizing it (Fig. 5). I would like to draw your attention to a couple of remarks opposing this argument:

1. It is true that human vision is not completely robust (*e.g.* to extreme rotation, scaling, or blur), but it is much more robust than our current systems, in particular in object recognition,

2. It takes a lot of skills and effort to create visual illusions, whereas finding adversarial examples for many images is strikingly easy – requiring a single step in the direction of the gradient,

3. The nature of visual illusions is very different from adversarial examples. Our visual system has evolved to enable us to function with high reliability without making deadly mistakes. It has sacrificed negligible accuracy over a very tiny sliver of visual space in order to gain immense robustness[11]. Adversarial examples,

---

[11]Just like we can not tell the exact temperature of water in degrees by placing our finger in it, our visual system can also not tell the exact intensity of a pixel. There has been no need for these feats through evolution. Instead, these systems operate by measuring relative difference and contrast. Visual system has chosen to pay a small cost (*e.g.* illusions), in order to become very robust to a large array of distortions and transformations. Also,

[10]*a.k.a* optical illusions.

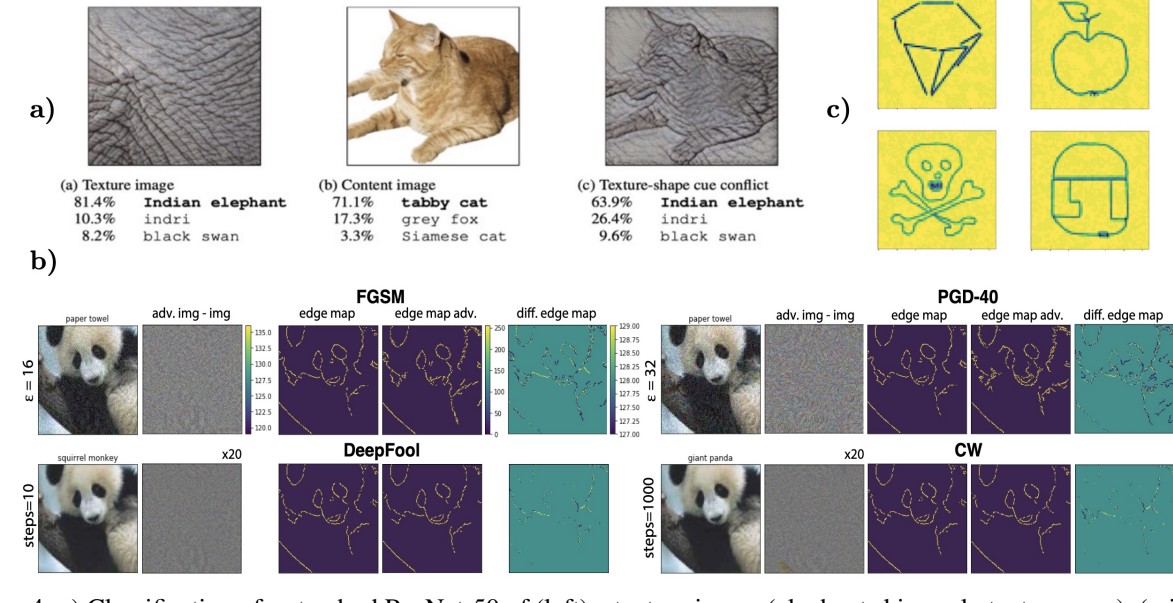

Figure 4: a) Classification of a standard ResNet-50 of (left) a texture image (elephant skin: only texture cues), (middle) a normal image of a cat (with both shape and texture cues), and (right) an image with a texture-shape cue conflict, generated by style transfer between the first two images (Geirhos et al., 2018a). b) Adversarial attacks against ResNet152 over the giant panda image using FGSM (Goodfellow et al., 2015), PGD-40 (Madry et al., 2017) ($\alpha$=8/255), DeepFool (Moosavi-Dezfooli et al., 2016) and Carlini-Wagner (Carlini & Wagner, 2017) attacks. Some columns in panel c show the difference ($\mathcal{L}_2$) between the original image (not shown) and the adversarial one (values shifted by 128 and clamped). The edge map (using Canny edge detector) remains almost intact at small perturbations. Notice that edges are better preserved for the PGD-40. Figure from (Borji, 2022). c) Samples from the Sketch dataset (Eitz et al., 2012) perturbed by FGSM $\epsilon = 8/255$ attack.

on the other hand, are prevalent and can cause catastrophic and costly mistakes. As such they are much more critical than visual illusions, and

4. Our visual system has a complicated web of interconnected regions. Different illusions target different functionalities of the brain (*e.g.* motion processing, attention and gaze, perceptual grouping, etc). Current vision systems are less complicated than our brains, and are limited in terms of the tasks they can perform[12]. It could well be that as models become more complicated, more adversarial examples can be synthesized for them.

A number of studies have conducted behavioral experiments to tap into the similarities of humans and DNNs in processing adversarial images. Elsayed *et al.* (Elsayed et al., 2018) found that **perceptible but class-preserving perturbations** that fool multiple machine learning models also fool time-limited humans. They state "adversarial examples are not (as is commonly misunderstood) defined to be imperceptible. If this were the case, it would be impossible by definition to make adversarial examples for humans, because changing the human's classification would constitute a change in what the human perceives". Their results are intriguing. There are, however, some concerns. First, there is a discrepancy between the above-mentioned definition and the formulation of the problem used in the literature (Eq. 1). Results are less surprising if the perturbation is allowed to be perceptible[13]. Second, the effects are statistically significant, but their size is small. Third, additional control experiments are needed to ensure these results can not be explained by other types of image distortions (*e.g.* blur). Finally, some follow-up works have reported contradictory results pertaining to agreement between humans and DNNs in interpreting adversarial images (*e.g.* (Zhou & Firestone, 2019; Dujmović et al., 2020)). Therefore, additional replication studies are required to draw strong conclusions.

Overall, the confident classification of the adversarial images by DNNs and the fact that a very small fraction of them can fool human subjects suggest that humans and DNNs perform image classification in fundamentally different ways.

---

humans are bad at recognizing objects in negative images or in images for which RGB channels have been shuffled, but are these serious shortcomings?! Perhaps, we should do the same in deep learning but the question is where and how to make a compromise.

[12]They are often built to do one task at a time.

[13]One may be able to increase the perturbation to a point where people make the desired mistakes!

Figure 5: Sample visual illusions. The top right image simultaneously depicts a portrait of a young lady or an old lady. In this example, the artist has carefully added features to make the portrait look like an old lady, while the new additions will not negatively impact the look of the young lady too much. For example, the right eyebrow of the old lady (marked in red below) does not distort the ear of the young lady too much. Creating images to fool the human visual system takes a lot of effort and special skills. Deep models are relatively much easier to fool. The middle image in the second row is a blend of Albert Einstein with Merlin Monroe (a hybrid image). Try to see the image from different distances, or squint your eyes. The bottom right image is an adversarial image generated by Elsayed *et al.* (Elsayed et al., 2018) to fool time-limited humans (making a cat look like a dog).

### 2.7. What can human vision offer?

Findings from visual neuroscience have contributed to expanding the deep learning toolbox. Further excavation is likely to add even more tools (Kruger et al., 2012). Some areas to look for inspiration are discussed below. Notice that there is much more to learn from the brain than those mentioned here (*e.g.* different types of normalization such as contrast normalization or divisive normalization (Heeger, 1992)).

The visual cortex has various types of connections including, short- and long-range horizontal connections as well as feedback connections[14]. These connections provide information to solve a range of tasks such as boundary ownership, figure-ground segmentation, contour tracing, perceptual grouping, as well as visual recognition. See (Kreiman & Serre, 2020; Serre, 2019) for reviews. Feedback signals may be the most critical missing piece in current DNNs.

Attention-based models, in particular Transformers (Vaswani et al., 2017; Dosovitskiy et al., 2020), have been very successful in several domains. They are also becoming the go-to models in computer vision. The

---

[14]Feedback connections are more prevalent in the brain than feed-forward ones.

attentional mechanisms incorporated in them, however, remain rather limited in comparison to the rich and diverse array of processes used by our visual system. Unlike CNNs, humans recognize objects one at a time through attention (Itti & Koch, 2001; Borji & Itti, 2012). Gaze and eye movements, in addition to providing computational efficiency via selecting and relaying the most important and relevant information to higher cortical areas for high-level processing, may also be crucial to gaining generalization across different tasks.

DNNs are frequently described as the best current models of biological vision. They have been utilized to predict the behavior of humans and non-human primates, large-scale activation of brain regions, as well as the firing patterns of individual neurons (see (Cichy et al., 2016; Cadieu et al., 2014; Kriegeskorte, 2015; Yamins & DiCarlo, 2016; Peterson et al., 2018; Kubilius et al., 2016)). There are, however, stark contrasts between the two systems. Utilizing minimal recognizable images, Ullman *et al.* (Ullman et al., 2016) argued that the human visual system uses features and processes that are not used by the current DNNs. Unlike human vision, DNNs are hindered drastically in recognizing objects in crowded scenes (Volokitin et al., 2017), and in detecting out-of-context objects (Rosenfeld et al., 2018). Models that can detect more robust features, are likely to better explain neuro-physiological and behavioral data. However, relying only on robust features is not enough. A robust model should also refrain from using features that are diagnostic of object category but are irrelevant to human vision (*i.e.* non-robust features).

## 3. Conclusion and path forward

Current DNNs have achieved impressive results over a wide variety of tasks and benchmarks. They are, however, very brittle. Current research on adversarial robustness has been trying hard to fix the problem, but so far no major success has been achieved. While continuing the current path, we should also be mindful of the bigger picture, which is building models that are truly general and robust. Here, I tried to shed some light on the problem by discussing it in a broader context.

Instead of fixing the existing architectures, for example by adding additional components to them, we may be better off inventing significantly different, or entirely new, classes of models. This may be a long and tedious path, but may yield bigger gains in the long run. We should also learn from human vision. One direction is to build models that are sensitive to the features used by the human visual system, while being insensitive to non-robust but highly predictive features.

Adversarial vulnerability of DNNs is very puzzling espe-

cially because it relates to our own viusal perception. This problem is where human and computer vision are highly entangled and as such provides a unique opportunity for cross-collaboration between the two fields, and for understanding the problem of vision in general.

## 4. Alternative Views

While we advocate for broadening the research focus to include perceptible perturbations and various image transformations, it is essential to recognize the foundational importance of addressing imperceptible adversarial attacks. These subtle perturbations pose significant security risks, especially in applications where inputs can be manipulated without detection, such as digital authentication systems or automated financial transactions. By ensuring models are robust against these nuanced attacks, we build a critical defense layer that protects against covert adversarial strategies.

Moreover, the study of imperceptible perturbations has led to valuable insights into the vulnerabilities of neural networks, shedding light on the intricate decision boundaries and feature sensitivities within these models. This understanding is crucial for the development of more resilient architectures. While expanding research to encompass a wider range of perturbations and transformations is beneficial, it should complement rather than replace the ongoing efforts to mitigate imperceptible adversarial attacks. A balanced approach ensures comprehensive robustness in machine learning models, addressing both subtle and overt adversarial challenges.

While expanding the scope of adversarial research to include perceptible attacks is vital, the value of work on imperceptible perturbations should not be diminished. It forms the theoretical and practical foundation that enables progress in tackling a broader spectrum of adversarial challenges.

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
