# OpenReview forum: "Position: Is Current Research on Adversarial Robustness Addressing the Right Problem?"
_ICML.cc/2025/Position_Paper_Track — Submitted to ICML 2025 Position Paper Track_

### Official Review · Reviewer_EHRo · 2025-02-24

**Significance:** 1
**Argument Clarity:** 1
**Rating:** 1
**Confidence:** 3

**Questions:**

Regarding (2) and (3), what happens when x and/or x' are in an area of the input space where no natural looking images exist? In that case, the error (disagreement of f and g) is perhaps less relevant/irrelevant. The footnote touches on this in suggesting that when g is not confident, things are different. But this point is not adequately explored. Given that legitimate inputs are on a submanifold of the huge R^d input space, then I am not sure if having an oracle g defined everywhere is well defined/makes sense. And certainly comparison with it is not sensible at all points.

Re equation (2), it is symmetric in x and x', so that if x' is an adv sample for x, then x is for x'. This is not necessarily wrong, but feels odd. Perhaps there needs to be a constraint re x being a legitimate image?? (whatever that may mean!)

Does (2) also need the condition that f(x)=g(x), so that x' is really an adversarial example and it is not just that f is wrong at and near x?

"Robust features will help alleviate backdoor attacks". Why? Is there a citation for this?

It is mentioned that adversarial robustness and generalisation  are two sides of the same coin. But surely by equation (3), if that is being put forward as a robustness measure, you are almost identifying them (same side of coin) as you are requiring f to generalise from its training data to agree with some (mythical) oracle precisely everywhere as a definition of robustness. But this seems to be a measure of generalisation success from the training data too.

Is it clear, as is claimed, that current neural network architectures cannot solve the general problem (assuming it could be well-posed)? ie since it can be argued that it follows from the universal approximation theorem that it should be possible (though resource limits may be a factor).

Is there a citation for the statement that the human visual system has sacrificed negligible accuracy in order to gain robustness?

**Discussion Potential:**

1

**Paper Summary:**

The paper suggests that there is a current focus on addressing only imperceptible adversarial perturbations, and that this is too narrow a focus. Rather, the paper suggests, we need to broaden the problem and discover model architectures robust to a class of perceptible adversarial transformations as well. It is stated that, at least for vision systems, learning from the human visual system may be a way forward in terms of utilising robust features and being insensitive to highly-predictive but non-robust features.

**Position:**

Yes

**Position In Title:**

No

**Related Work:**

1

**Strengths And Weaknesses:**

Overall, though the topic of the paper is clearly of some importance in the adversarial ML space, the development of the argument of the paper is not clear. There are a number of concerns detailed below, which, taken together, lead to the rating given. Overall, the paper I think does not give a precise definition of a broader class of perturbations in order to back up the key thrust, is too focused on vision systems and has unclear arguments in places. Details follow:

The position is unclear in the abstract - "short answer: yes. Long answer: no".

Ironically, for a paper suggesting current research is of too narrow a focus, the paper itself is focused almost exclusively on vision systems. As such, it omits discussion of application areas in which it may be argued that the focus on imperceptible perturbations is less of a concern. For example, in malware detection, adversarial examples are constrained only by maintaining the malicious functionality of the malware, and no strong constraint on the perturbation size (edit distance) exists.

One of the key statements is that "errors induced by perceptible perturbations are equally as important as errors induced by imperceptible perturbations". I honestly don't think this is too controversial, and is commonly recognised and understood - perhaps only the relative importance is open for serious debate. And of course the small perturbation problem is necessary (though not sufficient) to solve. But this is informed by the hardness of the problem, as well as defining exactly what an imperceptible perturbation is. Hence probably low impact/significance and little debate will ensue.

It is noted in sec 2.3 that imperceptibility and small perturbations in Lp norm are not equivalent. Yet Figure 2 literally identifies them as equivalent with the label and the suggested interpretation of the graphs.

Some queries re the definition (23) are captured in the Questions below.

Adversarial training is described as an ML "trick". Yet there has been evidence in the past that it effectively makes robust features more relevant - surely precisely what the paper is arguing for.

In the caption for figure 2, it is stated that a model is fully robust if it performs perfectly for all perturbation magnitudes. But what if it is in a region where the image is ambiguous? e.g. picture of cat and dog mixed in proportion theta to 1-theta. For theta close to 0.5, this is surely ambiguous and nothing can be inferred from agreement/disagreement with an "oracle". Hence it seems that the curves are not relevant for all theta, and so comparison of them across the range, as is done here, is meaningless.

I am not sure the Alternative Views section really proposes an Alternative View (ie that we ARE addressing the right problem). It argues a middle (reasonable!) ground of advocating complementing efforts to mitigate imperceptible attacks with addressing a wider range of perturbations/transformations. I think that is just a reasonable compromise/middle ground, not particularly controversial.

Minor comments:
- "y" above equation (1) is not defined/mentioned earlier. It is explained below (1), but that is too late for the informal statement above (1) to make sense.
- the reference in the text for Figure 1b suggests it shows natural looking images. It doesn't really.
- In 2.1, it is suggested that having a person in the loop for a face verification system defeats the point of automation. On the contrary, a security guard can be a defense against simple attempts at fooling (e.g. holding a picture of someone else in front of your face! :-) )
- theta (page 3) is introduced without any definition/discussion regarding its range of values, what exactly it is parameterizing, etc
- the point of the paper is to move beyond small perturbations, yet in section 2.6 point 2 it is argued that the existence  and ease of crafting of such small perturbations as evidence of a distinction between human and machine systems. Hence addressing robustness against small perturbations may remove this distinction - and so it may be a valid research direction after all.
- "Merlin" instead of "Marilyn" in Figure 5 caption
- Half of the images in Figure 5 are not referenced in the caption or text. They should not be there.
- The last sentences of section 2.7 are confusing. It says that relying on robust features is not enough. Then says that a robust model should not rely on non-robust features.

**Support:**

2

---

### Official Review · Reviewer_LnSX · 2025-02-28

**Significance:** 2
**Argument Clarity:** 2
**Rating:** 2
**Confidence:** 5

**Questions:**

Please refer to the weaknesses listed in Section “Strengths and Weaknesses”.

**Discussion Potential:**

1

**Paper Summary:**

This paper states the position regarding the research direction in terms of adversarial robustness. Specifically, the paper states that the research on adversarial robustness should not be limited to imperceptible perturbations. In contrary, the research should put more effort on the perceptible perturbations, such as geometric transformations and image distortions. Some discussions and literature reviews are presented.

## update after rebuttal

No rebuttal was posted.

**Position:**

Yes

**Position In Title:**

No

**Related Work:**

2

**Strengths And Weaknesses:**

Strengths:
1) This paper focuses on an important topic, i.e., the adversarial robustness of neural networks. This topic is widely studied in safety-critical scenarios.
2) The paper is easy to read and understand.

Weaknesses:
1) The position of this paper is not a novel direction in the field of adversarial robustness. Specifically, the adversarial attack which generates perceptible perturbation has been widely studied, such as the physical adversarial attacks which generate physical adversarial examples. As a result, the position of this paper is not new or impressive.
2) The position is not clearly stated in the abstract. The meanings of short-term goals and bigger gains are not clear.
3) The author states that “robust features will also help alleviate backdoor attacks”. However, this claim is not well supported. At least some references or empirical results should be provided.
4) I suggest the author to give a clear definition regarding the robust features, instead of just giving some examples to explain the meaning of robust features.

**Support:**

2

---

### Official Review · Reviewer_zShS · 2025-03-04

**Significance:** 3
**Argument Clarity:** 3
**Rating:** 3
**Confidence:** 3

**Questions:**

1. Authors argue that the bounded perturbation assumption is restrictive and misleading. How would authors propose to formally define and evaluate robustness in the expanded framework? What metrics would replace the current $\ell_p$-norm constraints?

2. Imperceptible perturbations may be a main concern in some sensitive areas, how do authors reconcile the position with this?

**Discussion Potential:**

3

**Paper Summary:**

This position paper challenges the current framework of adversarial robustness research. The authors argue that while research on imperceptible adversarial perturbations has yielded valuable insights, the current problem formulation is too narrow and potentially misleading.

The paper's key position is that focusing exclusively on imperceptible perturbations constrained by $\ell_p$-norms has created an artificially restrictive research setting that may be hindering broader progress toward truly robust models. Instead, the authors advocate for expanding the scope to address a more general robustness problem: developing architectures that are simultaneously robust to perceptible perturbations, geometric transformations (rotation, scaling), image distortions (lighting, blur), and other challenges like occlusion.

**Position:**

Yes

**Position In Title:**

Yes

**Related Work:**

2

**Strengths And Weaknesses:**

**Strengths:** The paper presents a clearly articulated and thoughtful critique of the current adversarial robustness research. The paper uses effective figures and examples to illustrate its points, particularly in demonstrating the limitations of $\ell_p$-norms as a measure of perceptual similarity and showing how different types of perturbations affect images.  The paper makes thoughtful connections to human vision research, providing biological inspiration for potential new directions in adversarial robustness research.

**Weaknesses:** While the paper provides a valuable high-level critique, it doesn't offer detailed technical proposals for how to actually implement the broader robustness paradigm it advocates. Some recent work that has begun to address the paper's concerns (e.g., research on broader distribution shifts, the robustness of image tasks under weather changes) could be more thoroughly discussed. The paper suggests abandoning the focus on imperceptible perturbations, but how to clearly define many different types of perturbations (which one is important)?

**Support:**

3

---

### Official Review · Reviewer_4FXF · 2025-03-13

**Significance:** 2
**Argument Clarity:** 2
**Rating:** 1
**Confidence:** 4

**Questions:**

What is $g$ in Eq. (2)?

**Discussion Potential:**

2

**Paper Summary:**

The paper criticizes about the current $\ell_p$ based adversarial robustness research and provides several related discussions as alternative research directions. Key points include: (a) $\ell_p$ threat models will be not enough to solve real-world adversarial threats; (b) research on perceptible perturbations, e.g., geometric transformations, image distortions, and occlusions, can be an alternative; (c) research needs to study more on human vision that rather focuses on shape, edges, and foreground objects rather than texture; (d) need for studying in the context of adversarial robustness vs. generalization; and finally remarking on that (e) we may need for new architectures to ultimately solve adversarial ML.

**Position:**

Yes

**Position In Title:**

No

**Related Work:**

1

**Strengths And Weaknesses:**

**Strengths**

- The paper is overall easy to follow.
- The points made in this paper are still valid and important research directions.
- The discussion about human visual illusion was interesting to me.

**Weakness**

- I generally feel the significant lack of discussion about recent related works, and that the overall points are a bit outdated; there is no “really new” research directions that are worth to be shared to ICML 2025 community in my opinion. For example, the paper does not cite paper after 2022 at all, which means that many research effort that are indeed aligned with the discussed topics here are dismissed.
- I think the points could be further strengthen by including discussion about recent impossibility results on adversarial machine learning; e.g., [1, 2, 3].
- The paper could benefit from discussing about adversarial threats of more advanced, modern deep learning systems beyond ResNets.
- The overall formatting and presentation can be improved. To name a few: (a) L15 - “Goodfellow et al. (Goodfellow et al., 2015)"; (b) Figures are in low resolution; (c) footnotes before comma; etc.
- The paper generally lacks of empirical supports, or limited elementary experiments, e.g., on MNIST and a single panda image.

[1] Gilmer et al., The Relationship Between High-Dimensional Geometry and Adversarial Examples, 2018.

[2] Mahloujifar et al., Empirically Measuring Concentration: Fundamental Limits on Intrinsic Robustness, NeurIPS 2019.

[3] Tramer et al., Fundamental Tradeoffs between Invariance and Sensitivity to Adversarial Perturbations, ICML 2020.

**Support:**

2

---

### Decision · Program_Chairs · 2025-04-27

**Decision:**

Reject

**Comment:**

The paper challenges the mainstream practice in adversarial robustness [for computer vision applications], which focuses on attacks involving imperceptible perturbations. While the topic is certainly important and timely, most reviewers felt that the paper missed the point in many aspects (quoting the reviewers' words):

- lack of discussion about recent related works
- the position defend by the paper is unclear (e.g.  "short answer: yes. Long answer: no" in the abstract)
- the paper itself is focused almost exclusively on vision systems
- the position of this paper is not a novel direction in the field of adversarial robustness

The authors did not submit a rebuttal to respond to these comments.